# ON LEARNING UNIVERSAL REPRESENTATIONS ACROSS LANGUAGES

**Xiangpeng Wei**[1,2]\*, **Rongxiang Weng**[3], **Yue Hu**[1,2], **Luxi Xing**[1,2], **Heng Yu**[3], **Weihua Luo**[3]

[1]Institute of Information Engineering, Chinese Academy of Sciences, Beijing, China
[2]School of Cyber Security, University of Chinese Academy of Sciences, Beijing, China
`{weixiangpeng,huyue,xingluxi}@iie.ac.cn`
[3]Machine Intelligence Technology Lab, Alibaba Group, Hangzhou, China
`{wengrx,yuheng.yh,weihua.luowh}@alibaba-inc.com`

## ABSTRACT

Recent studies have demonstrated the overwhelming advantage of cross-lingual pre-trained models (PTMs), such as multilingual BERT and XLM, on cross-lingual NLP tasks. However, existing approaches essentially capture the co-occurrence among tokens through involving the masked language model (MLM) objective with token-level cross entropy. In this work, we extend these approaches to learn sentence-level representations and show the effectiveness on cross-lingual understanding and generation. Specifically, we propose a **Hi**erarchical **Cont**rastive **L**earning (HICTL) method to (1) learn universal representations for parallel sentences distributed in one or multiple languages and (2) distinguish the semantically-related words from a shared cross-lingual vocabulary for each sentence. We conduct evaluations on two challenging cross-lingual tasks, XTREME and machine translation. Experimental results show that the HICTL outperforms the state-of-the-art XLM-R by an absolute gain of 4.2% accuracy on the XTREME benchmark as well as achieves substantial improvements on both of the high-resource and low-resource English→X translation tasks over strong baselines.

## 1 INTRODUCTION

Pre-trained models (PTMs) like ELMo (Peters et al., 2018), GPT (Radford et al., 2018) and BERT (Devlin et al., 2019) have shown remarkable success of effectively transferring knowledge learned from large-scale unlabeled data to downstream NLP tasks, such as text classification (Socher et al., 2013) and natural language inference (Bowman et al., 2015; Williams et al., 2018), with limited or no training data. To extend such *pretraining-finetuning* paradigm to multiple languages, some endeavors such as multilingual BERT (Devlin et al., 2019) and XLM (Conneau & Lample, 2019) have been made for learning cross-lingual representation. More recently, Conneau et al. (2020) present XLM-R to study the effects of training unsupervised cross-lingual representations at a huge scale and demonstrate promising progress on cross-lingual tasks.

However, all of these studies only perform a masked language model (MLM) with token-level (i.e., *subword*) cross entropy, which limits PTMs to capture the co-occurrence among tokens and consequently fail to understand the whole sentence. It leads to two major shortcomings for current cross-lingual PTMs, i.e., *the acquisition of sentence-level representations* and *semantic alignments among parallel sentences in different languages*. Considering the former, Devlin et al. (2019) introduced the next sentence prediction (NSP) task to distinguish whether two input sentences are continuous segments from the training corpus. However, this simple binary classification task is not enough to model sentence-level representations (Joshi et al., 2020; Yang et al., 2019; Liu et al., 2019; Lan et al., 2020; Conneau et al., 2020). For the latter, (Huang et al., 2019) defined the cross-lingual paraphrase classification task, which concatenates two sentences from different languages as input

---

\*Work done at Alibaba Group. Yue Hu and Heng Yu are the co-corresponding authors. We also made an official submission to XTREME (`https://sites.research.google/xtreme`), with several improved techniques used in (Fang et al., 2020; Luo et al., 2020).

and classifies whether they are with the same meaning. This task learns patterns of sentence-pairs well but fails to distinguish the exact meaning of each sentence.

In response to these problems, we propose to strengthen PTMs through learning universal representations among semantically-equivalent sentences distributed in different languages. We introduce a novel **Hi**erarchical **Con**trastive **L**earning (HICTL) framework to learn language invariant sentence representations via self-supervised non-parametric instance discrimination. Specifically, we use a BERT-style model to encode two sentences separately, and the representation of the first token (e.g., `[CLS]` in BERT) will be treated as the sentence representation. Then, we conduct instance-wise comparison at both sentence-level and word-level, which are complementary to each other. At the sentence level, we maximize the similarity between two parallel sentences while minimizing which among non-parallel ones. At the word-level, we maintain a bag-of-words for each sentence-pair, each word in which is considered as a positive sample while the rest words in vocabulary are negative ones. To reduce the space of negative samples, we conduct negative sampling for word-level contrastive learning. With the HICTL framework, the PTMs are encouraged to learn language-agnostic representation, thereby bridging the semantic discrepancy among cross-lingual sentences.

The HICTL is conducted on the basis of XLM-R (Conneau et al., 2020) and experiments are performed on several challenging cross-lingual tasks: language understanding tasks (e.g., XNLI, XQuAD, and MLQA) in the XTREME (Hu et al., 2020) benchmark, and machine translation in the IWSLT and WMT benchmarks. Extensive empirical evidence demonstrates that our approach can achieve consistent improvements over baselines on various tasks of both cross-lingual language understanding and generation. In more detail, our HICTL obtains absolute gains of 4.2% (up to 6.0% on zero-shot sentence retrieval tasks, e.g. BUCC and Tatoeba) accuracy on XTREME over XLM-R. For machine translation, our HICTL achieves substantial improvements over baselines on both low-resource (IWSLT English→X) and high-resource (WMT English→X) translation tasks.

## 2 RELATED WORK

**Pre-trained Language Models.** Recently, substantial work has shown that pre-trained models (PTMs) (Peters et al., 2018; Radford et al., 2018; Devlin et al., 2019) on the large corpus are beneficial for downstream NLP tasks. The application scheme is to fine-tune the pre-trained model using the limited labeled data of specific target tasks. For cross-lingual pre-training, both Devlin et al. (2019) and Conneau & Lample (2019) trained a transformer-based model on multilingual Wikipedia which covers various languages, while XLM-R (Conneau et al., 2020) studied the effects of training unsupervised cross-lingual representations on a very large scale.

For sequence-to-sequence pre-training, UniLM (Dong et al., 2019) fine-tuned BERT with an ensemble of masks, which employs a shared Transformer network and utilizing specific self-attention mask to control what context the prediction conditions on. Song et al. (2019) extended BERT-style models by jointly training the encoder-decoder framework. XLNet (Yang et al., 2019) trained by predicting masked tokens auto-regressively in a permuted order, which allows predictions to condition on both left and right context. Raffel et al. (2019) unified every NLP problem as a text-to-text problem and pre-trained a denoising sequence-to-sequence model at scale. Concurrently, BART (Lewis et al., 2020) pre-trained a denoising sequence-to-sequence model, in which spans are masked from the input but the complete output is auto-regressively predicted.

Previous works have explored using pre-trained models to improve text generation, such as pre-training both the encoder and decoder on several languages (Song et al., 2019; Conneau & Lample, 2019; Raffel et al., 2019) or using pre-trained models to initialize encoders (Edunov et al., 2019; Zhang et al., 2019a; Guo et al., 2020). Zhu et al. (2020) and Weng et al. (2020) proposed a BERT-fused NMT model, in which the representations from BERT are treated as context and fed into all layers of both the encoder and decoder. Zhong et al. (2020) formulated the extractive summarization task as a semantic text matching problem and proposed a Siamese-BERT architecture to compute the similarity between the source document and the candidate summary, which leverages the pre-trained BERT in a Siamese network structure. Our approach also belongs to the contextual pre-training so it could be applied to various downstream NLU and NLG tasks.

**Contrastive Learning.** Contrastive learning (CTL) (Saunshi et al., 2019) aims at maximizing the similarity between the encoded query $q$ and its matched key $k^+$ while keeping randomly sampled

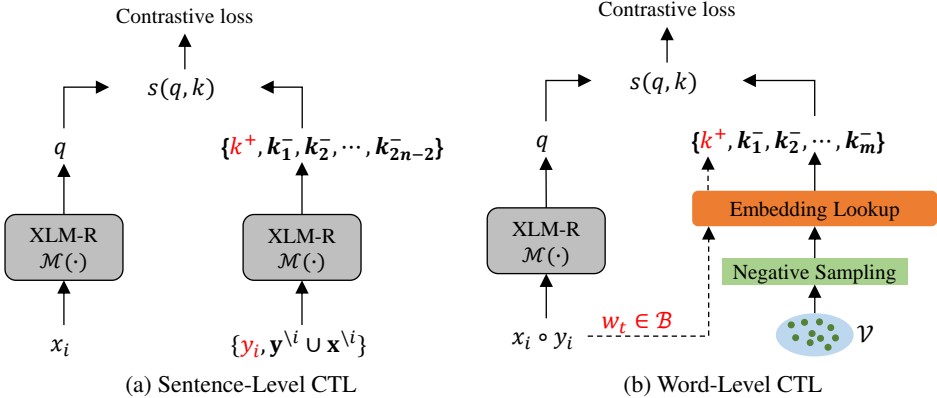

Figure 1: Illustration of **Hi**erarchical **C**ontrastive **L**earning (HICTL). $n$ is the batch size, $m$ denotes the number of negative samples for word-level contrastive learning. $\mathcal{B}$ and $\mathcal{V}$ indicates the bag-of-words of the instance $\langle x_i, y_i \rangle$ and the overall vocabulary of all languages, respectively.

keys $\{k_0^-, k_1^-, k_2^-, ...\}$ faraway from it. With similarity measured by a score function $s(q, k)$, a form of a contrastive loss function, called InfoNCE (Oord et al., 2018), is considered in this paper:

$$\mathcal{L}_{ctl} = -\log \frac{\exp(s(q, k^+))}{\exp(s(q, k^+)) + \sum_i \exp(s(q, k_i^-))}, \tag{1}$$

where the score function $s(q, k)$ is essentially implemented as the cosine similarity $\frac{q^T k}{\|q\| \cdot \|k\|}$. $q$ and $k$ are often encoded by a learnable neural encoder, such as BERT (Devlin et al., 2019) or ResNet (He et al., 2016). $k^+$ and $k^-$ are typically called positive and negative samples. In addition to the form illustrated in Eq. (1), contrastive losses can also be based on other forms, such as margin-based loses (Hadsell et al., 2006) and variants of NCE losses (Mnih & Kavukcuoglu, 2013).

Contrastive learning is at the core of several recent work on unsupervised or self-supervised learning from computer vision (Wu et al., 2018; Oord et al., 2018; Ye et al., 2019; He et al., 2019; Chen et al., 2020; Tian et al., 2020) to natural language processing (Mikolov et al., 2013; Mnih & Kavukcuoglu, 2013; Devlin et al., 2019; Clark et al., 2020b; Feng et al., 2020; Chi et al., 2020). Kong et al. (2020) improved language representation learning by maximizing the mutual information between a masked sentence representation and local n-gram spans. Clark et al. (2020b) utilized a discriminator to predict whether a token is replaced by a generator given its surrounding context. Iter et al. (2020) proposed to pre-train language models with contrastive sentence objectives that predict the surrounding sentences given an anchor sentence. In this paper, we propose HICTL to encourage parallel cross-lingual sentences to have the identical semantic representation and distinguish whether a word is contained in them as well, which can naturally improve the capability of cross-lingual understanding and generation for PTMs.

## 3 METHODOLOGY

### 3.1 HIERARCHICAL CONTRASTIVE LEARNING

We propose hierarchical contrastive learning (HICTL), a novel comparison learning framework that unifies cross-lingual sentences as well as related words. HICTL can learn from both non-parallel and parallel multilingual data, and the overall architecture of HICTL is illustrated in Figure 1. We represent a *training batch* of the original sentences as $\mathbf{x} = \{x_1, x_2, ..., x_n\}$ and its aligned counterpart is denoted as $\mathbf{y} = \{y_1, y_2, ..., y_n\}$, where $n$ is the batch size. For each pair $\langle x_i, y_i \rangle$, $y_i$ is either the translation in the other language of $x_i$ when using parallel data or the perturbation through reordering tokens in $x_i$ when only monolingual data is available. $\mathbf{x}^{\backslash i}$ is denoted as a modified version of $\mathbf{x}$ where the $i$-th instance is removed.

**Sentence-Level CTL.** As illustrated in Figure 1a, we apply the XLM-R as the encoder to represent sentences into hidden representations. The first token of every sequence is always a special token

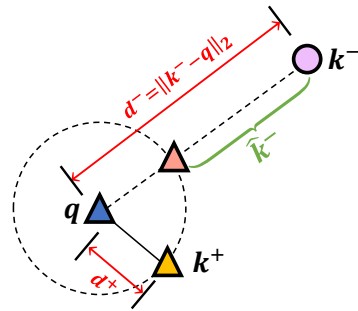

Figure 2: Illustration of constructing **h**ard **n**egative **s**amples (HNS). A circle (the radius is $d^+ = \| k^+ - q \|_2$) in the embedding space represents a manifold near in which sentences are semantically equivalent. We can generate a coherent sample (i.e., $\hat{k}^-$) that interpolate between known pair $q$ and $k^-$. The synthetic negative $\hat{k}^-$ can be controlled adaptively with proper difficulty during training. The curly brace in green indicates the walking range of hard negative samples, the closer to the circle the harder the sample is.

(e.g., `[CLS]`), and the final hidden state corresponding to this token is used as the aggregate sentence representation for pre-training, that is, $r_x = f \circ g(\mathcal{M}(x))$ where $g(\cdot)$ is the aggregate function and $f(\cdot)$ is a linear projection, $\circ$ denotes the composition of operations. To obtain universal representation among semantically-equivalent sentences, we encourage $r_{x_i}$ (the query, denoted as $q$) to be as similar as possible to $r_{y_i}$ (the positive sample, denoted as $k^+$) but dissimilar to all other instances (i.e., $\mathbf{y}^{\backslash i} \cup \mathbf{x}^{\backslash i}$, considered as a series of negative samples, denoted as $\{k_1^-, k_2^-, ..., k_{2n-2}^-\}$) in a training batch. Formally, the sentence-level contrastive loss for $x_i$ is defined as

$$\mathcal{L}_{sctl}(x_i) = - \log \frac{\exp \circ s(q, k^+)}{\exp \circ s(q, k^+) + \sum_{j=1}^{|\mathbf{y}^{\backslash i} \cup \mathbf{x}^{\backslash i}|} \exp \circ s(q, k_j^-)}. \tag{2}$$

Symmetrically, we also expect $r_{y_i}$ (the query, denoted as $\tilde{q}$) to be as similar as possible to $r_{x_i}$ (the positive sample, denoted as $\tilde{k}^+$) but dissimilar to all other instances in the same training batch, thus,

$$\mathcal{L}_{sctl}(y_i) = - \log \frac{\exp \circ s(\tilde{q}, \tilde{k}^+)}{\exp \circ s(\tilde{q}, \tilde{k}^+) + \sum_{j=1}^{|\mathbf{y}^{\backslash i} \cup \mathbf{x}^{\backslash i}|} \exp \circ s(\tilde{q}, \tilde{k}_j^-)}. \tag{3}$$

The sentence-level contrastive loss over the training batch can be formulated as

$$\mathcal{L}_S = \frac{1}{2n} \sum_{i=1}^n \left\{ \mathcal{L}_{sctl}(x_i) + \mathcal{L}_{sctl}(y_i) \right\}. \tag{4}$$

For sentence-level contrastive learning, we treat other instances contained in the training batch as negative samples for the current instance. However, such randomly selected negative samples are often uninformative, which poses a challenge of distinguishing very similar but nonequivalent samples. To address this issue, we employ smoothed linear interpolation (Bowman et al., 2016; Zheng et al., 2019) between sentences in the embedding space to alleviate the lack of informative samples for pre-training, as shown in Figure 2. Given a training batch $\{\langle x_i, y_i \rangle\}_{i=1}^n$, where $n$ is the batch size. In this context, having obtained the embeddings of a triplet, an anchor $q$ and a positive $k^+$ as well as a negative $k^-$ (supposing $q$, $k^+$ and $k^-$ are representations of sentences $x_i$, $y_i$ and $y_i^- \in \mathbf{x}^{\backslash i} \cup \mathbf{y}^{\backslash i}$, respectively), we construct a harder negative sample $\hat{k}^-$ to replace $k_j^-$:

$$\hat{k}^- = \begin{cases} q + \boldsymbol{\lambda}(k^- - q), \boldsymbol{\lambda} \in (\frac{d^+}{d^-}, 1] & if \quad d^- > d^+; \\ k^- & if \quad d^- \le d^+. \end{cases} \tag{5}$$

where $d^+ = \| k^+ - q \|_2$ and $d^- = \| k^- - q \|_2$. For the first condition, the hardness of $\hat{k}^-$ increases when $\boldsymbol{\lambda}$ becomes smaller. To this end, we intuitively set $\boldsymbol{\lambda}$ as

$$\boldsymbol{\lambda} = \left( \frac{d^+}{d^-} \right)^{\zeta \cdot p_{avg}^+}, \quad \zeta \in (0, 1) \tag{6}$$

where $p_{avg}^{+} = \frac{1}{100} \sum_{j \in [-100, -1]} e^{-\mathcal{L}_S^{(j)}}$ is the average log-probability over the last 100 training batches and $\mathcal{L}_S$ formulated in Eq. (4) is the sentence-level contrastive loss of one training batch. During pre-training, when the model tends to distinguish positive samples easily, which means negative samples are not informative already. At this time, $p_{avg}^{+} \uparrow$ and $\frac{d^+}{d^-} \downarrow$, which leads $\boldsymbol{\lambda} \downarrow$ and harder negative samples are adaptively synthesized in the following training steps, vice versa. As hard negative samples usually result in significant changes of the model parameters, we introduce the slack coefficient $\zeta$ to prevent the model from being trained in the wrong direction, when it accidentally switch from random negative samples to very hard ones. In practice, we empirically set $\zeta = 0.9$.

**Word-Level CTL.** Intuitively, predicting the related words in other languages for each sentence can bridge the representations of words in different languages. As shown in Figure 1b, we concatenate the sentence pair $\langle x_i, y_i \rangle$ as $x_i \circ y_i$: `[CLS]` $x_i$ `[SEP]` $y_i$ `[SEP]` and the bag-of-words of which is denoted as $\mathcal{B}$. For word-level contrastive learning, the final state of the first token is treated as the query ($\bar{q}$), each word $w_t \in \mathcal{B}$ is considered as the positive sample and all the other words ($\mathcal{V} \backslash \mathcal{B}$, i.e., the words in $\mathcal{V}$ that are not in $\mathcal{B}$ where $\mathcal{V}$ indicates the overall vocabulary of all languages) are negative samples. As the vocabulary usually with large space, we propose to only use a subset $\mathcal{S} \subset \mathcal{V} \backslash \mathcal{B}$ sampled according to the normalized similarities between $\bar{q}$ and the embeddings of the words. As a result, the subset $\mathcal{S}$ naturally contains the hard negative samples which are beneficial for learning high-quality representations (Ye et al., 2019). Specifically, the word-level contrastive loss for $\langle x_i, y_i \rangle$ is defined as

$$\mathcal{L}_{wctl}(x_i, y_i) = -\frac{1}{|\mathcal{B}|} \sum_{t=1}^{|\mathcal{B}|} \log \frac{\exp \circ s(\bar{q}, e(w_t))}{\exp \circ s(\bar{q}, e(w_t)) + \sum_{w_j \in \mathcal{S}} \exp \circ s(\bar{q}, e(w_j))}. \tag{7}$$

where $e(\cdot)$ is the embedding lookup function and $|\mathcal{B}|$ is the number of unique words in the concatenated sequence $x_i \circ y_i$. The overall word-level contrastive loss can be formulated as:

$$\mathcal{L}_W = \frac{1}{n} \sum_{i=1}^{n} \mathcal{L}_{wctl}(x_i, y_i). \tag{8}$$

**Multi-Task Pre-training.** Both MLM and translation language model (TLM) are combined with HICTL by default, as the prior work (Conneau & Lample, 2019) has verified the effectiveness of them in XLM. In summary, the model can be optimized by minimizing the entire training loss:

$$\mathcal{L} = \mathcal{L}_{LM} + \mathcal{L}_S + \mathcal{L}_W, \tag{9}$$

where $\mathcal{L}_{LM}$ is implemented as either the TLM when using parallel data or the MLM when only monolingual data is available to recover the original words of masked positions given the contexts.

## 3.2 CROSS-LINGUAL FINE-TUNING

**Language Understanding.** The representations produced by HICTL can be used in several ways for language understanding tasks whether they involve single text or text pairs. Concretely, ($i$) the `[CLS]` representation of single-sentence in sentiment analysis or sentence pairs in paraphrasing and entailment is fed into an extra output-layer for classification. ($ii$) The pre-trained encoder can be used to assign POS tags to each word or to locate and classify all the named entities in the sentence for structured prediction, as well as ($iii$) to extract answer spans for question answering.

**Language Generation.** We also explore using HICTL to improve machine translation. In the previous work, Conneau & Lample (2019) has shown that the pre-trained encoders can provide a better initialization of both supervised and unsupervised NMT systems. Liu et al. (2020b) has shown that NMT models can be improved by incorporating pre-trained sequence-to-sequence models on various language pairs but highest-resource settings. As illustrated in Figure 3, we use the model pre-trained by HICTL as the encoder, and add a new set of decoder parameters that are learned from scratch. To prevent pre-trained weights from being washed out by supervised training,

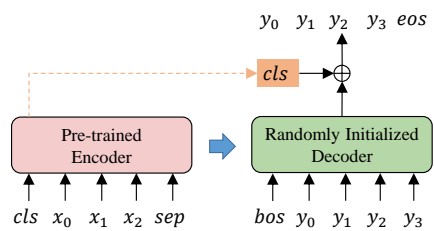

Figure 3: Fine-tuning on NMT task.

Table 1: Overall results on XTREME benchmark. Results of mBERT (Devlin et al., 2019), XLM (Conneau & Lample, 2019) and XLM-R (Conneau et al., 2020) are from XTREME (Hu et al., 2020). Results of ‡ are from our in-house replication. HNS is short for "**H**ard **N**egative **S**amples".

| Model | Pair sentence | | Structured prediction | | Question answering | | | Sentence retrieval | | |
|---|---|---|---|---|---|---|---|---|---|---|
| | XNLI | PAWS-X | POS | NER | XQuAD | MLQA | TyDiQA-GoldP | BUCC | Tatoeba | Avg. |
| Metrics | Acc. | Acc. | F1 | F1 | F1 / EM | F1 / EM | F1 / EM | F1 | Acc. | |
| *Cross-lingual zero-shot transfer (models are trained on English data)* | | | | | | | | | | |
| mBERT | 65.4 | 81.9 | 70.3 | 62.2 | 64.5 / 49.4 | 61.4 / 44.2 | 59.7 / 43.9 | 56.7 | 38.7 | 59.6 |
| XLM | 69.1 | 80.9 | 70.1 | 61.2 | 59.8 / 44.3 | 48.5 / 32.6 | 43.6 / 29.1 | 56.8 | 32.6 | 55.5 |
| XLM-R$_{Base}$ | 76.2 | - | - | - | - | 63.7 / 46.3 | - | - | - | - |
| HICTL$_{Base}$ | 77.3 | 84.5 | 71.4 | 64.1 | 73.5 / 58.7 | 65.8 / 47.6 | 61.9 / 42.8 | - | - | - |
| XLM-R | 79.2 | 86.4 | 73.8 | 65.4 | 76.6 / 60.8 | 71.6 / 53.2 | 65.1 / 45.0 | 66.0 | 57.3 | 68.2 |
| **HICTL** | **81.0** | **87.5** | **74.8** | **66.2** | **77.9 / 61.7** | **72.8 / 54.5** | **66.0 / 45.7** | **68.4** | **59.7** | **69.6** |
| *Translate-train-all (models are trained on English training data and its translated data on the target language)* | | | | | | | | | | |
| mBERT | 75.1 | 88.9 | - | - | 72.4 / 58.3 | 67.6 / 49.8 | 64.2 / 49.3 | - | - | - |
| XLM-R$^{‡}$ | 82.9 | 90.1 | 74.6 | 66.8 | 80.4 / 65.6 | 72.4 / 54.7 | 66.2 / 48.2 | 67.9 | 59.1 | 70.6 |
| HICTL | 84.5 | 92.2 | 76.8 | 68.4 | 82.8 / 67.3 | 74.4 / 57.1 | 69.7 / 52.5 | 71.8 | 63.1 | 73.2 |
| + HNS | **84.7** | **92.8** | **77.2** | **69.0** | **82.9 / 67.4** | **74.8 / 57.3** | **71.1 / 53.2** | **77.6** | **69.1** | **74.8** |

we train the encoder-decoder model in two steps. In the first step, we freeze the pre-trained encoder and only update the decoder. In the second step, we train all parameters for a relatively small number of iterations. In both cases, we compute the similarities between the `[CLS]` representation of the encoder and all target words in advance. Then we aggregate them with the logits before the softmax of each decoder step through an element-wise additive operation. The encoder-decoder model is optimized by maximizing the log-likelihood of bitext at both steps.

## 4 EXPERIMENTS

We consider two evaluation benchmarks: nine cross-lingual language understanding tasks in the XTREME benchmark and machine translation tasks (IWSLT'14 English↔German, IWSLT'14 English→Spanish, WMT'16 Romanian→English, IWSLT'17 English→{French, Chinese} and WMT'14 English→{German, French}). In this section, we describe the data and training details, and provide detailed evaluation results.

### 4.1 DATA AND MODEL

During pre-training, we follow Conneau et al. (2020) to build a Common-Crawl Corpus using the CCNet (Wenzek et al., 2019) tool[1] for monolingual texts. Table 7 (see appendix A) reports the language codes and data size in our work. For parallel data, we use the same (*English-to-X*) MT dataset as (Conneau & Lample, 2019), which are collected from MultiUN (Eisele & Yu, 2010) for French, Spanish, Arabic and Chinese, the IIT Bombay corpus (Kunchukuttan et al., 2018a) for Hindi, the OpenSubtitles 2018 for Turkish, Vietnamese and Thai, the EUbookshop corpus for German, Greek and Bulgarian, Tanzil for both Urdu and Swahili, and GlobalVoices for Swahili. Table 8 (see appendix A) shows the statistics of the parallel data.

We adopt the Transformer-Encoder (Vaswani et al., 2017) as the backbone with 12 layers and 768 hidden units for HICTL$_{Base}$, and 24 layers and 1024 hidden units for HICTL. We initialize the parameters of HICTL with XLM-R (Conneau et al., 2020). Hyperparameters for pre-training and fine-tuning are shown in Table 9 (see appendix B). We run the pre-training experiments on 8 V100 GPUs, batch size 1024. The number of negative samples $m$=512 for word-level contrastive learning.

### 4.2 EXPERIMENTAL EVALUATION

**Cross-lingual Language Understanding (XTREME)** There are nine tasks in XTREME that can be grouped into four categories: (*i*) sentence classification consists of Cross-lingual Natural Language Inference (XNLI) (Conneau et al., 2018) and Cross-lingual Paraphrase Adversaries from

---

[1]https://github.com/facebookresearch/cc_net

Table 2: Comparison with existing methods on XTREME tasks.

| Model | Pair sentence | | Structured prediction | | Question answering | | |
| | XNLI | PAWS-X | POS | NER | XQuAD | MLQA | TyDiQA-GoldP |
|---|---|---|---|---|---|---|---|
| Metrics | Acc. | Acc. | F1 | F1 | F1 / EM | F1 / EM | F1 / EM |
| *Translate-train-all* | | | | | | | |
| FILTER | 83.9 | 91.4 | 76.2 | 67.7 | 82.4 / **68.0** | **76.2** / 57.7 | 68.3 / 50.9 |
| VECO | 83.0 | 91.1 | 75.1 | 65.7 | 79.9 / 66.3 | 73.1 / 54.9 | **75.0** / **58.9** |
| **HICTL** | **84.7** | **92.8** | **77.2** | **69.0** | **82.9** / 67.4 | 74.8 / 57.3 | 71.1 / 53.2 |

Table 3: Ablation study on XTREME tasks.

| Model | XNLI Acc. | PAWS-X Acc. | POS F1 | NER F1 | XQuAD F1 / EM | MLQA F1 / EM | TyDiQA-GoldP F1 / EM | BUCC F1 | Tatoeba Acc. | Avg. |
|---|---|---|---|---|---|---|---|---|---|---|
| FULL MODEL | **84.7** | **92.8** | **77.2** | **69.0** | **82.9 / 67.4** | **74.8 / 57.3** | **71.1 / 53.2** | **77.6** | **69.1** | **74.8** |
| w/o Sentence-CTL | 82.9 | 90.5 | 75.9 | 67.8 | 82.3 / 66.7 | 74.3 / 56.5 | 69.7 / 52.3 | 71.4 | 62.6 | 72.4 |
| w/o Word-CTL | 84.3 | 92.1 | 76.3 | 68.4 | 82.5 / 66.9 | 74.1 / 56.7 | 70.2 / 52.5 | 76.8 | 68.4 | 74.2 |
| w/o MT data | 84.2 | 92.4 | 76.6 | 68.2 | 82.6 / 67.0 | 74.5 / 56.8 | 70.1 / 52.3 | 74.7 | 66.8 | 73.8 |

Word Scrambling (PAWS-X) (Zhang et al., 2019b). (*ii*) Structured prediction includes POS tagging and NER. We use POS tagging data from the Universal Dependencies v2.5 (Nivre et al., 2018) treebanks. Each word is assigned one of 17 universal POS tags. For NER, we use the Wikiann dataset (Pan et al., 2017). (*iii*) Question answering includes three tasks: Cross-lingual Question Answering (XQuAD) (Artetxe et al., 2019), Multilingual Question Answering (MLQA) (Lewis et al., 2019), and the gold passage version of the Typologically Diverse Question Answering dataset (TyDiQA-GoldP) (Clark et al., 2020a). (*iv*) Sentence retrieval includes two tasks: BUCC (Zweigenbaum et al., 2017) and Tatoeba (Artetxe & Schwenk, 2019), which aims to extract parallel sentences between the English corpus and target languages. As XTREME provides no training data, thus we directly evaluate pre-trained models on test sets.

Table 1 provides detailed results on four categories in XTREME. First, compared to the state of the art XLM-R baseline, HICTL further achieves significant gains of 1.43% and 2.80% on average on nine tasks with *cross-lingual zero-shot transfer* and *translate-train-all* settings, respectively. Second, mining hard negative samples via smoothed linear interpolation play an important role in contrastive learning, which significantly improves accuracy by 1.6 points on average. Third, HICTL with hardness aware augmentation delivers large improvements on zero-shot sentence retrieval tasks (scores 5.8 and 6.0 points higher on BUCC and Tatoeba, respectively). Following (Hu et al., 2020), we directly evaluate pre-trained models on test sets without any extra labeled data or fine-tuning techniques used in (Fang et al., 2020; Luo et al., 2020). These results demonstrate the capacity of HICTL on learning cross-lingual representations. We also compare our best model with two existing models: FILTER (Fang et al., 2020) and VECO (Luo et al., 2020). The results demonstrate that HICTL achieves the best performance on most tasks with less monolingual data.

Ablation experiments are present at Table 3. Comparing the full model, we can draw several conclusions: (1) removing the sentence-level CTL objective hurts performance consistently and significantly, (2) the word-level CTL objective has least drop compared to others, and (3) the parallel (MT) data has a large impact on zero-shot multilingual sentence retrieval tasks. Moreover, Table 2 provides the comparisons between HICTL and existing methods.

**Machine Translation**   The main idea of HICTL is to summarize cross-lingual parallel sentences into a shared representation that we term as semantic embedding, using which semantically related words can be distinguished from others. Thus it is natural to apply this global embedding to text generation. We fine-tune the pre-trained HICTL with the base setting on machine translation tasks with both low-resource and high-resource settings. For the low-resource scenario, we choose IWSLT'14 English↔German (En↔De)[2], IWSLT'14 English→Spanish (En→Es), WMT'16

---

[2]We split 7k sentence pairs from the training dataset for validation and concatenate dev2010, dev2012, tst2010, tst2011, tst2012 as the test set.

Table 4: **BLEU scores [%] on high-resource tasks.** Results with † and ‡ are from VECO (Luo et al., 2020) and our in-house implementation, respectively. In our implementation, we use XLM-R and the best version of HiCTL (pre-traind with CCNet-100 and hard negative samples) to initialize the encoder, respectively.

| Model | Layers | | WMT'14 | |
|---|---|---|---|---|
| | Encoder | Decoder | En→De | En→Fr |
| *Randomly Initialize* | | | | |
| Transformer-Big (Vaswani et al., 2017) | 6 | 6 | 28.4 | 41.0 |
| Deep-Transformer (Liu et al., 2020a) | 60 | 12 | 30.1 | 43.8 |
| Deep MSC Model (Wei et al., 2020) | 18 | 6 | 30.56 | - |
| *Pre-trained Models Initialize* | | | | |
| CTNMT (Yang et al., 2020) | 18 | 6 | 30.1 | 42.3 |
| BERT-fused NMT (Zhu et al., 2020) | 18 | 6 | 30.75 | 43.78 |
| mBART† (Liu et al., 2020b) | 12 | 12 | 30.0 | 43.2 |
| VECO (Luo et al., 2020) | 24 | 6 | 31.5 | **44.4** |
| XLM-R‡ | 24 | 6 | 30.91 | 43.27 |
| HiCTL | 24 | 6 | **31.74** | 43.95 |

Table 5: **BLEU scores [%] on low-resource tasks.** Results with ‡ are from our in-house implementation. We provide additional experimental results (to follow experiments in Zhu et al. (2020)) on IWSLT'14 English→Spanish (En→Es) task. HiCTL$_{Base}$ represents the BASE sized model that is pre-trained on CCNet-100 with hard negative samples.

| Model | IWSLT'14 | | | WMT'16 | IWSLT'17 | |
|---|---|---|---|---|---|---|
| | En→De | De→En | En→Es | Ro→En | En→Fr | En→Zh |
| Transformer (Vaswani et al., 2017)‡ | 28.64 | 34.51 | 39.3 | 33.51 | 35.8 | 26.5 |
| BERT-fused NMT (Zhu et al., 2020) | 30.45 | 36.11 | 41.4 | 39.10 | 38.7 | 28.2 |
| HiCTL$_{Base}$ | **31.88** | **37.96** | **42.1** | **39.88** | **40.2** | **29.9** |

Romanian→English (Ro→En), IWSLT'17 English→French (En→Fr) and English→Chinese (En→Zh) translation[3]. There are 160k, 183k, 236k, 235k, 0.6M bilingual sentence pairs for En↔De, En→Es, En→Fr, En→Zh and Ro→En tasks. For the rich-resource scenario, we work on WMT'14 En→{De, Fr}, the corpus sizes are 4.5M and 36M respectively. We concatenate *newstest 2012* and *newstest 2013* as the validation set and use *newstest 2014* as the test set.

During fine-tuning, we use the pre-trained model to initialize the encoder and introduce a randomly initialized decoder. We develop a shallower decoder with 4 identical layers to reduce the computation overhead. At the first fine-tune step, we concatenate the datasets of all language pairs in either low-resource or high-resource settings to optimize the decoder only until convergence[4]. Then we tune the whole encoder-decoder model using a per-language corpus at the second step. The initial learning rate is 2e-5 and `inverse_sqrt` learning rate (Vaswani et al., 2017) scheduler is also adopted. For WMT'14 En→De, we use beam search with width 4 and length penalty 0.6 for inference. For other tasks, we use width 5 and a length penalty of 1.0. We use `multi-bleu.perl` to evaluate IWSLT'14 En↔De and WMT tasks, but `sacreBLEU` for the remaining tasks, for fair comparison with previous work.

Results on both high-resource and low-resource tasks are reported in Table 4 and Table 5, respectively. We implemented standard Transformer (apply the `base` and `big` setting for IWSLT and WMT tasks respectively) as baseline. The proposed HiCTL can improve the BLEU scores of the eight tasks by 3.34, 2.95, 3.24, 3.45, 2.8, 6.37, 4.4, and 3.4. In addition, our approach also outperforms the BERT-fused model (Yang et al., 2020), a method treats BERT as an extra context

---

[3]https://wit3.fbk.eu/mt.php?release=2017-01-ted-test

[4]Zhao et al. (2020) conducted a theoretical investigation on learning universal representations for the task of multilingual MT, while we directly use a shared encoder and decoder across languages for simplicity.

Table 6: **BLEU scores [%] on Zero-shot MT via Language Transfer.** We bold the highest transferring score for each language family.

| Test Languages | Fine-tuning Languages | | | |
| | Cs→En | | Hi→En | |
| | mBART | HiCTL | mBART | HiCTL |
| --- | --- | --- | --- | --- |
| Cs→En | 21.6 | 22.4 | - | |
| Ro→En | **19.5** | 19.0 | - | |
| It→En | 16.7 | 18.6 | - | |
| Nl→En | 17.0 | 18.1 | - | |
| Hi→En | - | | 23.5 | 25.2 |
| Ne→En | - | | 14.5 | **16.0** |
| Si→En | - | | 13.0 | 14.7 |
| Gu→En | - | | 0.0 | 0.1 |

and fuses the representations extracted from BERT with each encoder and decoder layer. Note we achieve new state-of-the-art results on IWSLT'14 En→De, IWSLT'17 En→{Fr, Zh} translations. These improvements show that mapping different languages into a universal representation space is beneficial for both low-resource and high-resource translations.

We also evaluate our model on tasks where no bi-text is available for the target language pair. Following mBART (Liu et al., 2020b), we adopt the setting of language transfer. That is, no bi-text for the target pair is available, but there is bi-text for translating from some other language into the target language. For explanation, supposing there is no parallel data for the target language pair Italian→English (It→En), but we can transfer knowledge learned from Czech→English (Cs→En, a high-resource language pair) to It→En. We consider X→En translation, covering Indic languages (Ne, Hi, Si, Gu) and European languages (Ro, It, Cs, Nl). For European languages, we fine-tune on Cs→En translation, the parallel data is from WMT'19 that contains 11M sentence pairs. We test on {Cs, Ro, It, Nl}→En, in which test sets are from previous WMT (Cs, Ro) or IWSLT (It, Nl) competitions. For Indic languages, we fine-tune on Hi→En translation (1.56M sentence pairs are from IITB (Kunchukuttan et al., 2018b)), and test on {Ro, It, Cs, Nl}→En translations.

Results are shown in Table 6. We can always obtain reasonable transferring scores at low-resource pairs over different fine-tuned models. However, our experience shows that the randomly initialized models without pre-training always achieve near 0 BLEU. The underlying scenario is that multilingual pre-training produces universal representations across languages so that once the model learns to translate one language, it learns to translate all languages with similar representations. Moreover, a failure happened in Gu→En translation, we conjecture that we only use 0.3GB monolingual data for pre-training, which is difficult to learn informative representations for Gujarati.

## 5 CONCLUSION

We have demonstrated that pre-trained language models (PTMs) trained to learn commonsense knowledge from large-scale unlabeled data highly benefit from hierarchical contrastive learning (HICTL), both in terms of cross-lingual understanding and generation. Learning universal representations at both word-level and sentence-level bridges the semantic discrepancy across languages. As a result, our HICTL sets a new level of performance among cross-lingual PTMs, improving on the state of the art by a large margin.

## ACKNOWLEDGMENTS

We would like to thank the anonymous reviewers for the helpful comments. We also thank Jing Yu for the instructive suggestions. This work is supported by the National Key R&D Program of China under Grant No.2017YFB0803301 and No. 2018YFB1403202.

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

Table 7: The statistics of CCNet corpus used for pretraining.

| Code | Size (GB) | Code | Size (GB) | Code | Size (GB) | Code | Size (GB) | Code | Size (GB) |
|------|-----------|------|-----------|------|-----------|------|-----------|------|-----------|
| af | 1.3 | et | 6.1 | ja | 24.2 | mt | 0.2 | sq | 3.0 |
| am | 0.7 | eu | 2.0 | jv | 0.2 | my | 0.9 | sr | 5.1 |
| ar | 20.4 | fa | 21.6 | ka | 3.4 | ne | 2.6 | su | 0.1 |
| as | 0.1 | fi | 19.2 | kk | 2.6 | nl | 15.8 | sv | 10.8 |
| az | 3.6 | fr | 46.5 | km | 1.0 | no | 3.7 | sw | 1.6 |
| be | 3.5 | fy | 0.2 | kn | 1.2 | om | 0.1 | ta | 8.2 |
| bg | 22.6 | ga | 0.5 | ko | 17.2 | or | 0.6 | te | 2.6 |
| bn | 7.9 | gd | 0.1 | ku | 0.4 | pa | 0.8 | th | 14.7 |
| br | 0.1 | gl | 2.9 | ky | 1.2 | pl | 16.8 | tl | 0.8 |
| bs | 0.1 | gu | 0.3 | la | 2.5 | ps | 0.7 | tr | 17.3 |
| ca | 10.1 | ha | 0.3 | lo | 0.6 | pt | 15.9 | ug | 0.4 |
| cs | 16.3 | he | 6.7 | lt | 7.2 | ro | 8.6 | uk | 9.1 |
| cy | 0.8 | hi | 20.2 | lv | 6.4 | ru | 48.1 | ur | 5.0 |
| da | 15.2 | hr | 5.4 | mg | 0.2 | sa | 0.3 | uz | 0.7 |
| de | 46.3 | hu | 9.5 | mk | 1.9 | sd | 0.4 | vi | 44.6 |
| el | 29.3 | hy | 5.5 | ml | 4.3 | si | 2.1 | xh | 0.1 |
| en | 49.7 | id | 10.6 | mn | 1.7 | sk | 4.9 | yi | 0.3 |
| eo | 0.9 | is | 1.3 | mr | 1.3 | sl | 2.8 | zh | 36.8 |
| es | 44.6 | it | 19.8 | ms | 3.2 | so | 0.4 | - | - |

Table 8: Parallel data used for pre-training.

| Code | Sentence Pair (#millions) | Code | Sentence Pair (#millions) |
|------|---------------------------|------|---------------------------|
| en-ar | 9.8 | en-ru | 11.7 |
| en-bg | 0.6 | en-sw | 0.2 |
| en-de | 9.3 | en-th | 3.3 |
| en-el | 4.0 | en-tr | 0.5 |
| en-es | 11.4 | en-ur | 0.7 |
| en-fr | 13.2 | en-vi | 3.5 |
| en-hi | 1.6 | en-zh | 9.6 |

## A  PRE-TRAINING DATA

During pre-training, we follow Conneau et al. (2020) to build a Common-Crawl Corpus using the CCNet (Wenzek et al., 2019) tool[5] for monolingual texts. Table 7 reports the language codes and data size in our work. For parallel data, we use the same (*English-to-X*) MT dataset as (Conneau & Lample, 2019), which are collected from MultiUN (Eisele & Yu, 2010) for French, Spanish, Arabic and Chinese, the IIT Bombay corpus (Kunchukuttan et al., 2018a) for Hindi, the OpenSubtitles 2018 for Turkish, Vietnamese and Thai, the EUbookshop corpus for German, Greek and Bulgarian, Tanzil for both Urdu and Swahili, and GlobalVoices for Swahili. Table 8 shows the statistics of the parallel data.

## B  HYPERPARAMETERS FOR PRE-TRAINING AND FINE-TUNING

As shown in Table 9, we present the hyperparameters for pre-training HICTL. We use the same vocabulary as well as the sentence-piece model with XLM-R (Conneau et al., 2020). During fine-tuning on XTREME, we search the learning rate over {5e-6, 1e-5, 1.5e-5, 2e-5, 2.5e-5, 3e-5} and batch size over {16, 32} for BASE-size models. And we select the best LARGE-size model by searching the learning rate over {3e-6, 5e-6, 1e-5} as well as batch size over {32, 64}.

---

[5]https://github.com/facebookresearch/cc_net

Table 9: Hyperparameters used for pre-training.

| Hyperparameters | BASE | LARGE |
|---|---|---|
| Number of layers | 12 | 24 |
| Hidden size | 768 | 1024 |
| FFN inner hidden size | 3072 | 4096 |
| Attention heads | 12 | 16 |
| Mask percent (monolingual/bilingual) | 15%/25% | 15%/25% |
| Adam $\epsilon$ | 1e-6 | 1e-6 |
| Adam $\beta$ | (0.9, 0.98) | (0.9, 0.999) |
| Learning rate | 2.5e-4 | 1e-4 |
| Learning rate schedule | linear | linear |
| Warmup steps | 10,000 | 10,000 |
| Attention dropout | 0.1 | 0.1 |
| Dropout | 0.1 | 0.1 |
| Max sequence length (monolingual/bilingual) | 256 | 256 |
| Batch size | 1024 | 1024 |
| Training steps | 200k | 200k |

Table 10: Results on Cross-lingual Natural Language Inference (XNLI) for each language. We report the accuracy on each of the 15 XNLI languages and the average accuracy of our HICTL as well as five baselines: BiLSTM (Conneau et al., 2018), mBERT (Devlin et al., 2019), XLM (Conneau & Lample, 2019), Unicoder (Huang et al., 2019) and XLM-R (Conneau et al., 2020). Results of ‡ are from our in-house replication.

| MODEL | en | fr | es | de | el | bg | ru | tr | ar | vi | th | zh | hi | sw | ur | Avg |
|---|---|---|---|---|---|---|---|---|---|---|---|---|---|---|---|---|
| *Evaluation of cross-lingual sentence encoders (Cross-lingual transfer)* | | | | | | | | | | | | | | | | |
| BiLSTM | 73.7 | 67.7 | 68.7 | 67.7 | 68.9 | 67.9 | 65.4 | 64.2 | 64.8 | 66.4 | 64.1 | 65.8 | 64.1 | 55.7 | 58.4 | 65.6 |
| mBERT | 81.4 | - | 74.3 | 70.5 | - | - | - | - | 62.1 | - | - | 63.8 | - | - | 58.3 | - |
| XLM | 85.0 | 78.7 | 78.9 | 77.8 | 76.6 | 77.4 | 75.3 | 72.5 | 73.1 | 76.1 | 73.2 | 76.5 | 69.6 | 68.4 | 67.3 | 75.1 |
| Unicoder | 85.1 | 79.0 | 79.4 | 77.8 | 77.2 | 77.2 | 76.3 | 72.8 | 73.5 | 76.4 | 73.6 | 76.2 | 69.4 | 69.7 | 66.7 | 75.4 |
| XLM-R$_{Base}$ | 85.8 | 79.7 | 80.7 | 78.7 | 77.5 | 79.6 | 78.1 | 74.2 | 73.8 | 76.5 | 74.6 | 76.7 | 72.4 | 66.5 | 68.3 | 76.2 |
| HICTL$_{Base}$ | **86.3** | **80.5** | **81.3** | **79.5** | **78.9** | **80.6** | **79.0** | **75.4** | **74.8** | **77.4** | **75.7** | **77.6** | **73.1** | **69.9** | **69.7** | **77.3** |
| *Machine translate at training (Translate-train)* | | | | | | | | | | | | | | | | |
| BiLSTM | 73.7 | 68.3 | 68.8 | 66.5 | 66.4 | 67.4 | 66.5 | 64.5 | 65.8 | 66.0 | 62.8 | 67.0 | 62.1 | 58.2 | 56.6 | 65.4 |
| mBERT | 81.9 | - | 77.8 | 75.9 | - | - | - | - | 70.7 | - | - | 76.6 | - | - | 61.6 | - |
| XLM | 85.0 | 80.2 | 80.8 | 80.3 | 78.1 | 79.3 | 78.1 | 74.7 | 76.5 | 76.6 | 75.5 | 78.6 | 72.3 | 70.9 | 63.2 | 76.7 |
| Unicoder | 85.1 | 80.0 | 81.1 | 79.9 | 77.7 | 80.2 | 77.9 | 75.3 | 76.7 | 76.4 | 75.2 | 79.4 | 71.8 | 71.8 | 64.5 | 76.9 |
| HICTL$_{Base}$ | **85.7** | **81.3** | **82.1** | **80.2** | **81.4** | **81.0** | **80.5** | **79.7** | **77.4** | **78.2** | **77.5** | **80.2** | **75.4** | **73.5** | **72.9** | **79.1** |
| *Fine-tune multilingual model on all training sets (Translate-train-all)* | | | | | | | | | | | | | | | | |
| XLM | 85.0 | 80.8 | 81.3 | 80.3 | 79.1 | 80.9 | 78.3 | 75.6 | 77.6 | 78.5 | 76.0 | 79.5 | 72.9 | 72.8 | 68.5 | 77.8 |
| Unicoder | 85.6 | 81.1 | 82.3 | 80.9 | 79.5 | 81.4 | 79.7 | 76.8 | 78.2 | 77.9 | 77.1 | 80.5 | 73.4 | 73.8 | 69.6 | 78.5 |
| XLM-R$_{Base}$ | 85.4 | 81.4 | 82.2 | 80.3 | 80.4 | 81.3 | 79.7 | 78.6 | 77.3 | 79.7 | 77.9 | 80.2 | 76.1 | 73.1 | 73.0 | 79.1 |
| HICTL$_{Base}$ | 86.5 | 82.3 | 83.2 | 80.8 | 81.6 | 82.2 | 81.3 | 80.5 | 78.1 | 80.4 | 78.6 | 80.7 | 76.7 | 73.8 | 73.9 | 80.0 |
| XLM-R | 89.1 | 85.1 | 86.6 | 85.7 | 85.3 | 85.9 | 83.5 | 83.2 | 83.1 | **83.7** | 81.5 | 83.7 | **81.6** | 78.0 | **78.1** | 83.6 |
| XLM-R‡ | 88.9 | 84.7 | 86.2 | 84.8 | 85.0 | 85.3 | 82.4 | 82.7 | 82.4 | 82.8 | 80.9 | 83.0 | 80.2 | 77.3 | 77.2 | 82.9 |
| **HICTL** | **89.3** | **85.5** | **86.9** | **86.1** | **85.7** | **86.1** | **83.7** | **83.9** | **83.3** | 83.5 | **81.8** | **84.2** | 81.0 | **78.4** | 77.9 | **83.8** |

## C  RESULTS FOR EACH DATASET AND LANGUAGE

Below, we provide detailed results for each dataset and language on XTREME, as shown in Table 10-14. Results of XLM-R are from our implementation.

## D  VISUALIZATION OF SENTENCE EMBEDDINGS

We collect 10 sets of samples from WMT'14-19, each of them contains 100 parallel sentences distributed in 5 languages. As the t-SNE visualization in Figure 4, a set of sentences under the same meaning are clustered more densely for HICTL than XLM-R, which reveals the strong capability

Table 11: PAWS-X accuracy scores for each language.

| Model | en | de | es | fr | ja | ko | zh | avg |
|---|---|---|---|---|---|---|---|---|
| *Translate-train-all* | | | | | | | | |
| XLM-R | 95.7 | 92.2 | 92.7 | 92.5 | 84.7 | 85.9 | 87.1 | 90.1 |
| HICTL, Wiki-15 + MT | 96.6 | 93.2 | 93.3 | 92.9 | 86.5 | 87.3 | 88.6 | 91.2 |
| HICTL, CCNet-100 + MT | 96.9 | 93.8 | 94.4 | 94.3 | 88.0 | 88.2 | 89.4 | 92.2 |
| +HARD NEGATIVE SAMPLES | **97.4** | **94.2** | **95.0** | **94.2** | **89.1** | **89.5** | **90.2** | **92.8** |

Table 12: POS results (Accuracy) for each language.

| Model | af | ar | bg | de | el | en | es | et | eu | fa | fi | fr | he | hi | hu | id | it |
|---|---|---|---|---|---|---|---|---|---|---|---|---|---|---|---|---|---|
| *Translate-train-all* | | | | | | | | | | | | | | | | | |
| XLM-R | 90.6 | 67.4 | 89.1 | 89.9 | 86.8 | 96.3 | 89.6 | 87.1 | 74.0 | 70.8 | 86.0 | 87.7 | 68.6 | 77.4 | 82.8 | 72.6 | 91.1 |
| HICTL, Wiki-15 + MT | 91.0 | 69.3 | 89.1 | 89.4 | 87.8 | 97.6 | 88.2 | 88.2 | 74.8 | 72.0 | 86.7 | 87.9 | 70.2 | 79.0 | 84.2 | 74.3 | 90.8 |
| HICTL, CCNet-100 + MT | 91.8 | 70.2 | 90.7 | 90.8 | 89.0 | **98.3** | 89.7 | **90.1** | 76.2 | 73.0 | 88.5 | **90.2** | 70.7 | **80.0** | **86.4** | 74.5 | **92.0** |
| +HARD NEGATIVE SAMPLES | **92.2** | **71.0** | **91.5** | **91.3** | **90.0** | 97.7 | **91.0** | 89.4 | 75.7 | **73.5** | **88.8** | 90.1 | **71.1** | 79.7 | 85.4 | **75.1** | 91.7 |
| | ja | kk | ko | mr | nl | pt | ru | ta | te | th | tl | tr | ur | vi | yo | zh | avg |
| *Translate-train-all* | | | | | | | | | | | | | | | | | |
| XLM-R | 17.3 | 78.3 | 55.5 | 82.1 | 89.8 | 88.9 | 89.8 | 65.7 | 87.0 | 48.6 | 92.9 | 77.9 | 71.7 | 56.8 | 24.7 | 27.2 | 74.6 |
| HICTL, Wiki-15 + MT | 28.4 | 79.2 | 54.2 | 80.7 | 90.9 | 88.4 | 90.5 | 67.3 | 89.1 | 48.7 | 92.2 | 77.6 | 72.0 | 58.8 | 27.2 | 27.1 | 75.5 |
| HICTL, CCNet-100 + MT | 30.2 | 80.4 | 55.1 | 82.1 | 91.2 | 90.2 | 90.7 | 68.1 | 90.1 | 50.3 | **95.2** | 78.7 | 73.3 | 59.2 | 27.8 | 27.9 | 76.8 |
| +HARD NEGATIVE SAMPLES | **31.9** | **80.9** | **57.0** | **83.5** | **91.7** | **91.0** | **91.2** | **69.5** | **90.8** | 50.3 | 94.8 | **79.4** | **73.4** | **59.5** | **28.6** | **28.7** | **77.2** |

of HICTL on learning universal representations across different languages. Note that the t-SNE visualization of HICTL still demonstrates some noises, we attribute them to the lack of hard negative examples for sentence-level contrastive learning and leave this to future work for consideration.

Table 13: NER results (F1) for each language.

| Model | en | af | ar | bg | bn | de | el | es | et | eu | fa | fi | fr | he | hi | hu | id | it | ja | jv |
|---|---|---|---|---|---|---|---|---|---|---|---|---|---|---|---|---|---|---|---|---|
| *Translate-train-all* | | | | | | | | | | | | | | | | | | | | |
| XLM-R | 86.8 | 81.4 | 55.2 | 82.9 | 81.1 | 79.1 | 81.5 | 81.1 | 81.3 | 60.6 | 64.1 | 80.6 | 83.2 | 60.1 | 76.1 | 79.4 | 53.2 | 80.7 | 22.7 | 63.9 |
| HICTL, Wiki-15 + MT | 87.0 | **82.3** | 55.2 | 84.7 | 79.0 | 81.2 | 80.1 | 81.6 | 79.8 | 61.4 | 61.9 | **82.8** | 80.5 | 60.4 | 74.6 | 79.8 | 54.8 | 83.5 | 24.9 | **66.1** |
| HICTL, CCNet-100 + MT | 88.6 | 80.9 | 55.4 | **85.6** | 81.8 | 82.0 | 82.5 | 80.8 | 81.2 | 62.5 | 64.2 | 81.2 | **83.0** | 60.3 | **77.3** | **84.4** | 55.8 | 83.7 | 26.0 | 65.0 |
| +HARD NEGATIVE SAMPLES | **88.9** | 82.0 | **56.6** | 83.7 | **83.4** | **82.8** | **84.8** | **83.0** | **83.8** | **65.4** | **65.4** | 82.0 | 82.6 | **60.5** | 74.7 | 81.5 | **58.1** | **84.7** | **27.9** | 65.9 |
| | ka | kk | ko | ml | mr | ms | my | nl | pt | ru | sw | ta | te | th | tl | tr | ur | vi | yo | zh |
| XLMR | 74.2 | 58.0 | 63.3 | 68.3 | 69.8 | 59.5 | 57.5 | 86.2 | 82.3 | 68.5 | 70.7 | 59.8 | 58.5 | 2.4 | 72.6 | 75.9 | 59.7 | 79.4 | 37.0 | 35.4 |
| HICTL, Wiki-15 + MT | 75.0 | 56.7 | 62.2 | 69.4 | 68.8 | 57.9 | 55.6 | **87.9** | 84.2 | **71.9** | 74.4 | 61.6 | **59.2** | 2.2 | 74.2 | 79.5 | 58.1 | 83.0 | 35.2 | 33.0 |
| HICTL, CCNet-100 + MT | 72.8 | 57.6 | 64.6 | 70.4 | 71.5 | **61.1** | **59.0** | 87.7 | **85.1** | 70.3 | 74.3 | 60.6 | 57.9 | **5.6** | 77.5 | 79.0 | **59.8** | 83.7 | 37.7 | 36.9 |
| +HARD NEGATIVE SAMPLES | **76.8** | **60.9** | **65.0** | **71.4** | **72.5** | 59.0 | 56.3 | 85.9 | 84.5 | 71.4 | **75.6** | **62.9** | 58.8 | 3.9 | **77.7** | **80.4** | 59.1 | 83.6 | **37.7** | **37.2** |

Table 14: Tatoeba results (Accuracy) for each language

| Model | af | ar | bg | bn | de | el | es | et | eu | fa | fi | fr | he | hi | hu | id | it | ja |
|---|---|---|---|---|---|---|---|---|---|---|---|---|---|---|---|---|---|---|
| *Translate-train-all* | | | | | | | | | | | | | | | | | | |
| XLM-R | 59.7 | 50.5 | 72.2 | 45.4 | 89.5 | 61.3 | 77.6 | 51.7 | 38.6 | 71.7 | 72.8 | 76.9 | 66.3 | 73.1 | 65.1 | 77.5 | 68.5 | 63.1 |
| HICTL, Wiki-15 + MT | 61.5 | 51.4 | 76.1 | 47.9 | 92.1 | 63.4 | 80.5 | 55.9 | 37.8 | 74.6 | 76.7 | 78.0 | 68.4 | 74.5 | 68.8 | 80.4 | 70.2 | 63.9 |
| HICTL, CCNet-100 + MT | 63.0 | 50.9 | 76.8 | 47.0 | 94.6 | 68.8 | 80.9 | 59.3 | 41.5 | 77.3 | 78.2 | 80.3 | 70.2 | 77.9 | 72.1 | 81.3 | 73.7 | 66.2 |
| +HARD NEGATIVE SAMPLES | **68.9** | **57.7** | **83.2** | **55.4** | **98.2** | **74.5** | **88.5** | **62.4** | **47.7** | **80.2** | **82.9** | **85.5** | **79.1** | **85.0** | **76.8** | **90.3** | **80.8** | **72.7** |
| | jv | ka | kk | ko | ml | mr | nl | pt | ru | sw | ta | te | th | tl | tr | ur | vi | zh |
| XLM-R | 15.8 | 53.3 | 51.2 | 63.1 | 66.2 | 59.0 | 81.0 | 84.4 | 76.9 | 19.8 | 28.3 | 37.8 | 28.9 | 36.7 | 68.9 | 26.6 | 77.9 | 69.8 |
| HICTL, Wiki-15 + MT | 18.7 | 55.8 | 51.0 | 65.5 | 67.3 | 61.2 | 82.9 | 84.4 | 78.3 | 22.2 | 28.6 | 41.4 | 33.5 | 41.6 | 71.2 | 26.7 | 80.2 | 73.6 |
| HICTL, CCNet-100 + MT | 19.6 | 57.3 | 54.6 | 68.0 | 71.8 | 62.0 | 88.1 | 88.9 | 77.7 | 26.1 | 32.9 | 39.5 | 32.9 | 43.2 | 71.2 | 27.8 | 79.9 | 74.7 |
| +HARD NEGATIVE SAMPLES | **27.2** | **63.0** | **61.5** | **72.6** | **75.3** | **67.8** | **92.8** | **92.8** | **85.4** | **32.0** | **36.7** | **47.8** | **41.5** | **49.8** | **77.0** | **34.3** | **84.3** | **81.3** |

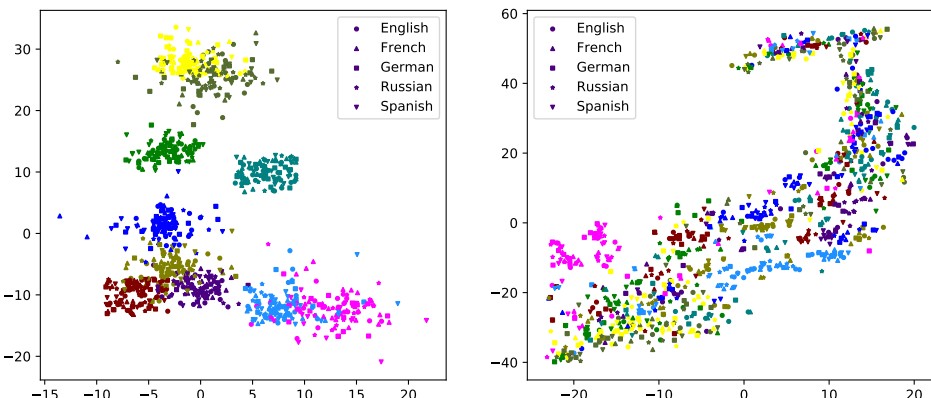

Figure 4: Visualizations (t-SNE projection) of sentence embeddings output by HICTL (*left*) and XLM-R (*right*). We collect 10 sets of samples from WMT'14-19, each of them contains 100 parallel sentences distributed in 5 languages (i.e., English, French, German, Russian, and Spanish). Each set is identified by a color and different languages marked by different shapes. We can see that a set of sentences under the same meaning are clustered more densely for HICTL than XLM-R, which reveals the strong capability of HICTL on learning universal representations across different languages.

