# OpenReview forum: "On Learning Universal Representations Across Languages"
_ICLR.cc/2021/Conference — ICLR 2021 Poster_

### Official Review · AnonReviewer4 · 2020-10-29
**Universal representations, at what cost?**

**Rating:** 7
**Confidence:** 4

**Review:**

Summary

The work applies and adjusts contrastive learning in the subject area of pre-training language models. The work first identifies the challenges with the current landscape of Masked Language Models with limits to learning sentence-level representations and semantic alignments in sentences of different languages. To take care of these gaps, the authors propose using HCTL as an approach that can learn more universal representations for sentences across different languages. The work builds on top of the BERT models, with the adjusted contrastive learning objective goal.

Strengths

The paper is well written. The authors work to clearly identify shortcomings in the current literature. The identification of contrastive learning as a possible approach to learn better representations.

Questions

- Using the [CLS} token has been one of the ways to capture the sentence embedding for BERT, correct? I do understand that there are other ways to try to extract sentence embedding from pre-trained BERT, but your approach is not new in this sense?

- I am not sure I missed it, but I do not have an objective view on the computational overhead (especially on having to sample the right-hand side of the non-similar keys). Simply, how much more time do I spend using this type of training as compared to the prior approaches?

- With the prior question, the results, show consistent improvements (your biggest strength along with the representation itself being comparable) but are a slight improvement in individual metrics (a point or two). Given the goal of developing universal cross-lingual representations, why is it that if we do learn these better representations are we not doing Very Very well?

- Might it be that the prior models, already have these representations already embedded in their models and your work just extracted them? Could there be another approach to XML-R for example, that extracts these representations?

- Thank you for including different sizes of corpora and also some low-resource languages. Kiswahili is the smallest parallel data that you have, and I wondered what would happen with a smaller language? For low-resource languages, it would be insightful to also have examples of where the failures happen and why.

- Please expand the future 3 captions. it is hard to understand what it represents from the caption. Just make sure the caption is completely descriptive. The figure is also very small and hard to read.

I think the contrastive learning approach is very interesting for this use-case and made my reading and evaluation of this paper more interesting. I look forward to the responses from the authors.

---

> ### Author Response · Authors · 2020-11-16
> **Thanks for your positive comments and very constructive suggestions**
>
> Thanks for your positive comments and very constructive suggestions! We address your concerns and questions as follows:
>
> 1.&ensp;Yes, the final state of the [CLS] token has been widely used to capture the sentence embedding in the NSP (next sentence prediction) task for BERT or in the SOP (sentence order prediction) task for ALBERT, which learns coherence between sentences. We have followed this general approach to encode sentences. There are other ways to aggregate sentences, such as the average on representations of all positions. We will keep working on this and hope to provide more direct experiments or analyses to have a deeper understanding of the principle in the next revision.
>
> 2.&ensp;We answer in two folds: (a) For sentence-level CTL, we use other instances contained in the training batch as randomly negative samples and mine hard ones via smoothed linear interpolation (see Appendix C) for current training instance. Those are easily obtained without extra sampling. (b) For word-level CTL, we only sample a subset of the whole vocabulary as negative samples (the number is 512 in our experiments), which has little impact on computational efficiency.
>
> 3.&ensp;Very good point. In the original paper, we are indeed not doing very very well. The reason is that we treat other instances contained in the training batch as negative samples for the current instance for sentence-level CTL. However, such randomly selected negative samples are often uninformative, which poses a challenge of distinguishing very similar but nonequivalent samples (as examples with noises shown in Figure 3). To address this issue, we employ smoothed linear interpolation between sentences in the embedding space to alleviate the lack of informative samples for pre-training. By doing so, our HiCTL achieves consistent and significant improvements on all tasks in XTREME. Details are reported in Appendix D of the updated version.
>
> 4.&ensp;Intuitively, the prior models should contain the representations that describe patterns of co-occurrence among tokens by performing a masked language model with token-level cross-entropy for pre-training. However, these representations can not be used to distinguish the exact meaning of each sentence or word. There are other approaches to represent sentences in embedding space, such as sentence-BERT, which uses the average of all positions or computes a max-over-time of the output states. We will make a combination between these strategies and our HiCTL.
>
> 5.&ensp;We have evaluated our HiCTL on zero-shot machine translation tasks via language transfer (see appendix G of the updated version for details). We experiment on two families of languages: Indic languages (Ne, Hi, Si, Gu) and European languages (Ro, It, Cs, Nl). For each family of languages, we fine-tune on the relative high-resource pairs (e.g., Hi$\rightarrow$En and Cs$\rightarrow$En) and directly test on the rest low/zero-resource pairs. From the results, we can observe reasonable transferring scores at all low/zero-resource pairs except Gujarati$\rightarrow$English. Our conjecture is that we use extremely little monolingual data (0.3GB) of Gujarati for pre-training, which is difficult to learn informative representations.
>
> 6.&ensp;We have enriched the caption of Figure 3 and enlarge it in Appendix H of the updated version.
>
> Thank you again for the constructive suggestions and we value these ideas to improve our paper. Hope these could address your concerns.
>
> Reimers et al. 2019. Sentence-BERT: Sentence Embeddings using Siamese BERT-Networks.

---

### Official Review · AnonReviewer3 · 2020-11-01
**extension of cross lingual pre-trained models with sentence- and word-level contrastive losses**

**Rating:** 5
**Confidence:** 4

**Review:**

The paper proposes a pre-trained language model variant which extends XLM-R (multilingual masked model) with two new objectives. The main difference to most other models is that the new losses are contrastive losses (however, as pointed out by the authors, other contrastive losses had been used before in e.g. ELECTRA). The first additional loss is a sentence-level one - where a [CLS] token is trained to be close to the positive sample, the paired sentence, with other sentences as negative samples. The same is done at word level, where the bag of words constructed from two sentences becomes the set of positive samples and other vocabulary words are negative samples.
Contrastive losses are promising and the paper shows positive results when adding them to the previously proposed XLM-R model. The review of previous work is thorough and very helpful to place the work proposed in the existing literature. However I had difficulties understanding the impact of the changes proposed and disentangling the different factors that may have led to the results. At the end of reading this paper, I am not sure if implementing what the authors proposed, versus other variations of existing models, would have given the same improvements: While these improvements can be seen across many data sets, they are often modest. The proposal does not offer any other advantages, such as computational efficiency.
For the NMT experiments, additional experiments on En-Es and En-Ro (to follow experiments in Zhu et al 2020), and/or back-translation experiments would have made the impact of the method clearer. Given that the main contribution of the paper is empirical (none of the ideas are new), better and more comprehensive experimental results would have strengthened this work.
The following are clarification questions/comments:
- The query and key terminology used in section 2 is confusing: why not use negative/positive sample notation from the Saunshi et al, 2019 and  Oord et al, 2018 papers? Section 3.1 introduces r_x, which is yet another notation for the query q.
- Figure 1: please clarify the notation used in the caption (e.g. the set B is defined only later, similarly n, m, V).
- The losses in equations (2) and (3) are symmetric: if the data pairs are symmetric, which seems to be the case, why distinguish between queries and keys at all and define two identical, but symmetric losses?
- First paragraph in Word-Level CTL in Section 3.1: This should be rephrased in order to clarify the motivation for the word level loss.
- I couldn't find details regarding the negative samples for the sentence loss: no of negative samples, how are they obtained, etc.

---

> ### Author Response · Authors · 2020-11-16
> **Thanks a lot for your insightful comments and constructive suggestions**
>
> Thanks a lot for your insightful comments and constructive suggestions! We have provided comprehensive experimental results as an appendix in the newly uploaded version according to your suggestions. We address your concerns and questions as follows:
>
> \## The impact of the proposed method \##
>
> 1.&ensp;The proposed HiCTL improves predominant PTMs in two folds: (a) The sentence-CTL encourages semantically equivalent sentences in different languages to have the universal representation, as visualized in Figure 3 (Page 8). (b) The word-CTL bridges the gap between word embeddings of different languages and encourages the model to discover related words given a semantic embedding. We have provided the ablations about only doing sentence-CTL or word-CTL to better understand how different factors affect the results in Appendix D (Table 10).
>
> 2.&ensp;Compared to other variations of existing models (i.e., [FILTER] and [VECO]), our HiCTL achieves substantial improvements on most tasks of the XTREME benchmark. All three methods used CCNet-100 corpus for pre-training, but our HiCTL differs in learning universal representations via sentence-level and word-level contrast that is novel for this use-case. Details are reported in Appendix D (Table 11) of the updated version.
>
> 3.&ensp;Our main contribution of the paper is to extend existing PTMs to learn universal representations across different languages and to show the effectiveness on cross-lingual understanding and generation tasks, rather than improving computational efficiency. HiCTL initializes from XLM-R and introduces no additional parameters, that is, HiCTL improves by a large margin without reducing computational efficiency.
>
> \## About additional experiments on NMT tasks \##
>
> Very good point. We have provided additional experiments on En-Es and Ro-En as well as back-translation experiments on Ro-En. On IWSLT’14 En-Es/WMT’16 Ro-En, our HiCTL leads to 0.7/0.8 BLEU improvements over the bert-fused model [Zhu et al, 2020], and outperforms standard Transformer as well as back-translation methods by a large margin. Details are reported in Appendix F (Table 17 and Table 18) of the new version.
>
> \## About clarifications \##
>
> 1.&ensp;Following MoCo [He et al., 2019], we use the notation $k^+$/$k^-$ to represent positive/negative sample, which is identical to [Saunshi et al, 2019; Oord et al, 2018]. Moreover, we introduce the notation $r_x$ to represent the embedding of the sentence $x$, rather than specific query or key. When a sentence $x$ is treated as the anchor (query), thus $q = r_x$, otherwise $k^+=r_x$ or $k^-=r_x$.
>
> 2.&ensp;We have clarified the notations used in the caption of Figure 1 according to your suggestions.
>
> 3.&ensp;A pair of sentences (i.e., $\langle x, y \rangle$) of each training instance are semantically equivalent but not the same one. We hope to learn a symmetric representation space, in which not only $x$ is most similar to $y$ but also $y$ is most similar to $x$ among all data points.
>
> 4.&ensp;We will rephrase the first paragraph in word-CTL (Section 3.1) in the next version to make the motivation of the word-CTL clearer.
>
> 5.&ensp;As pointed out by the original paper (Sentence-Level CTL, Section 3.1, Page 4), we treat other instances contained in the training batch as negative examples for current instance. Specifically, given a training batch {$(x_1, y_1), (x_2, y_2), …, (x_n, y_n)$}, for the query $x_i$, its paired counterpart $y_i$ is treated as the positive sample but other instances {$(x_1, y_1), …, (x_{i-1}, y_{i-1}), (x_{i+1}, y_{i+1}), …, (x_n, y_n)$} are treated as negative samples. Moreover, we also generate hard negative samples via linear interpolation, details can be seen in Appendix C and D.
>
> Thanks again and please consider our clarifications or discussions above and the updated experiments (see Appendix D and F). Hope these could address your concerns.
>
> [FILTER] Fang et al. FILTER: An Enhanced Fusion Method for Cross-lingual Language Understanding.
>
> [VECO] Luo et al. VECO: Variable Encoder-Decoder Pre-training for Cross-lingual Understanding and Generation.

---

### Official Review · AnonReviewer2 · 2020-11-02
**HICTL presents some interesting contrastive losses for representation learning**

**Rating:** 7
**Confidence:** 5

**Review:**

Summary:
The paper presents HICTL which enables models to learn sentence level representations and uses contrastive learning to force better language agnostic representations for large multilingual encoders. They obtain significant gains on the XTREME benchmark and also on standard MT benchmarks.

Reasons for score:
I score this paper a 6. The constrastive losses introduced in the paper are interesting. The sentence-level and word-level CTL definitely seem to have an impact on the downstream performance. The improvements on XTREME over XLM-R is pretty strong. However, the improvements in MT are not that convincing. I would have also liked to see more ablations. Finally, the authors initialize from XLM-R and fine-tune on 15 languages with both monolingual and parallel data. I would have liked to see this number (i.e. number of languages) be much larger.

Cons:
- Why did you only choose to fine-tune on the 15 languages from XNLI?
- Can you present the breakdown of results in different tasks by language as an appendix? Specifically, I would like to see if the improvements are only coming in the 15 languages you fine-tune on or on others as well. Do you have an answer for this?
- I would like to see ablations for XTREME where you only did sentence-CTL or only did word-CTL. Other interesting ablations without parallel data would have also added more value to the paper. Ablations of the amount of data used would also be interesting.
- Writing could have been clearer. I found a lot of grammar mistakes in the paper which need to be corrected.
- Why did you not make an official submission to the XTREME leaderboard? That would have been a more fair evaluation of your system.


Minor comments:
- Word-level CTL: change "are in two folds" to "are two fold".
- Table 2: It would be good to have an average column which will make it easier for the reader.

---

> ### Author Response · Authors · 2020-11-16
> **Thanks for the valuable comments and very inspiring suggestions**
>
> Thanks for the valuable comments and very inspiring suggestions! Due to time limits, we could only address major points in the first discussion stage, but we’ll make sure to reflect all advice in the second stage.
>
> \## Pre-training HiCTL on more Languages \##
>
> This is a good point. We have extended HiCTL to 100 languages with hard negative samples as Appendix D (Table 9) and uploaded a new version. Specifically,
>
> 1.&ensp;The proposed HiCTL benefits from more languages. Compared to the model pre-trained on the 15 languages from XNLI, the other averagely scores 1.3 points higher on the XTREME benchmark when it was pre-trained on CCNet corpus in 100 languages.
>
> 2.&ensp;Mining hard negative samples via smoothed linear interpolation plays an important role in sentence-level contrastive learning. Based on the model pre-trained on CCNet-100 corpus, it significantly improves accuracy by additional 1.6 points on average. It worth noting that, our best model outperforms XLM-R by 4.2 points on nine tasks in XTREME.
>
> 3.&ensp;HiCTL with hardness aware augmentation delivers large improvements on zero-shot sentence retrieval tasks, scoring 9.7 and 10.0 points higher than XLM-R on BUCC and Tatoeba, respectively. Following the [XTREME team], we directly evaluated pre-trained models on test sets without any extra labeled data or fine-tuning techniques in the current version.
>
> We will add these details to the main pages after the review period. In the future, we will make an official submission to the XTREME leaderboard with several improved techniques used in [FILTER] and [VECO] for a fairer comparison.
>
> \## Detailed results for each language \##
>
> We have provided detailed results for three tasks (i.e., POS, NER, and Tatoeba, each of which covers at least 33 languages) in the newly uploaded version. Details are reported in Appendix E. Whether pre-training with Wiki-15 or CCNet-100 corpus, the results show that the improvements of HiCTL are coming in almost all languages.
>
> \## About improvements in MT \##
>
> Thank you for bringing this up. The bilingual data of MT tasks can be naturally applied to the sentence-level contrastive learning, thus we chose to further pre-train HiCTL on IWSLT and WMT parallel corpora before fine-tuning on MT tasks. For a fairer comparison, we also replaced HiCTL with XLM-R that was also pre-trained on WMT and IWSLT tasks (see Table 5, Page 8). However, we agree that evaluating the proposed HiCTL without task-adaptive pre-training would be more convincing. We have revised the experiments on MT tasks (refer to Appendix F for details) and will add these details to the main pages after the review period.
>
> \## About ablations \##
>
> Very good point. Due to the time limits, we could not provide ablations about only doing sentence-CTL or word-CTL, as well as other ablations (e.g., without parallel data and the amount of data used for pre-training) in the first discussion stage, but we promise to add these details to the next version during stage two. We also appreciate the reviewer’s understanding that pre-training on the large HiCTL model is very time-consuming.
>
> \## About writing \##
>
> We have carefully revised the paper according to your suggestions.
>
> Thanks again for the constructive suggestions. Hope these could address your concerns.
>
> [XTREME team] Hu et al. XTREME: A Massively Multilingual Multi-task Benchmark for Evaluating Cross-lingual Generalization.
>
> [FILTER] Fang et al. FILTER: An Enhanced Fusion Method for Cross-lingual Language Understanding.
>
> [VECO] Luo et al. VECO: Variable Encoder-Decoder Pre-training for Cross-lingual Understanding and Generation.

---

> > ### Author Response · Authors · 2020-11-22
> > **Updated response about ablations**
> >
> > Thanks again for the useful comments. We have provided several ablations for XTREME (Appendix D, Table 10). Comparing the full model, we can draw several conclusions: (1) removing the sentence-level CTL objective hurts performance consistently and significantly, (2) the word-level CTL objective has least drop compared to others, and (3) the parallel (MT) data has a large impact on zero-shot multilingual sentence retrieval tasks.

---

> > > ### Comment · AnonReviewer2 · 2020-11-24
> > > **Thank you for the additional experiments**
> > >
> > > I would like to thank to authors for addressing my feedback and running various experiments which include:
> > > 1) Extending to 100 languages
> > > 2) Ablation study on the different objectives
> > > 3) Ablation without the parallel data
> > >
> > > They also provided the detailed per task per language breakdown on XTREME.
> > >
> > > These experiments definitely improve the paper further. I will increase my score from 6 to 7.

---

### Author Response · Authors · 2020-11-22
**Updated Version**

We want to thank the reviewers again for their time and the useful comments. We have updated the paper with a detailed appendix, the changes are as follows:
* We extended HiCTL to 100 languages with pre-training on CCNet-100 corpus and mining hard negative examples via smoothed linear interpolation.
* We conducted detailed ablations on only doing sentence-CTL or word-CTL, as well as pre-training without parallel data, to better understand how different factors affect the results.
* We provided comprehensive experimental results on high-resource, low-resource and zero-resource (with language transfer) machine translation.

---

### Comment · ~Carmen_Amo_Alonso1 · 2021-02-17
**MINOR COMMENT**

Hi
Please change "cab" to can in the last sentence in last page

---

### Comment · ~Abhyuday_Jagannatha4 · 2021-03-28
**Request for codes**

@Authors: could you please release the codes?
@ICLR: could you please retract the paper if author do not release the code? Since papers without code do not help the community and this is the best for everyone not to accept such papers, as one cannot even make sure if the results are really true. thank you.

---

> ### Author Response · Authors · 2021-03-29
> **Response**
>
> Thanks for your attention. We are sorting out the relevant data and documents, and will release the source codes as soon as possible.

---

### Decision · Program_Chairs · 2021-01-07
**Final Decision**

**Decision:**

Accept (Poster)

**Comment:**

This paper presents two new representation learning tasks (losses) based on contrastive learning that---when combined with a language modeling loss---result in a better multilingual model. Experiments on machine translation and XTREME demonstrate the benefit of the proposed method compared to strong baselines.

I think this is an interesting paper that advances multilingual representation learning. The authors have incorporated many suggestions from the reviewers to improve the paper during the rebuttal period. I recommend to accept the paper, but also strongly suggest the authors to make an official submission to XTREME to validate their results.